# Ribosomal RNA Modulates Aggregation of the *Podospora* Prion Protein HET-s

**DOI:** 10.3390/ijms21176340

**Published:** 2020-09-01

**Authors:** Yanhong Pang, Petar Kovachev, Suparna Sanyal

**Affiliations:** Department of Cell and Molecular Biology, Uppsala University, Box-596, Biomedical Center, 751 24 Uppsala, Sweden; pangyanhong1982@hotmail.com (Y.P.); peterkovachev@gmail.com (P.K.)

**Keywords:** ribosomal RNA, prion aggregation, *P. anserina* prion protein HET-s, PFAR

## Abstract

The role of the nucleic acids in prion aggregation/disaggregation is becoming more and more evident. Here, using HET-s prion from fungi *Podospora anserina* (*P. anserina*) as a model system, we studied the role of RNA, particularly of different domains of the ribosomal RNA (rRNA), in its aggregation process. Our results using Rayleigh light scattering, Thioflavin T (ThT) binding, transmission electron microscopy (TEM) and cross-seeding assay show that rRNA, in particular the domain V of the major rRNA from the large subunit of the ribosome, substantially prevents insoluble amyloid and amorphous aggregation of the HET-s prion in a concentration-dependent manner. Instead, it facilitates the formation of the soluble oligomeric “seeds”, which are capable of promoting de novo HET-s aggregation. The sites of interactions of the HET-s prion protein on domain V rRNA were identified by primer extension analysis followed by UV-crosslinking, which overlap with the sites previously identified for the protein-folding activity of the ribosome (PFAR). This study clarifies a missing link between the rRNA-based PFAR and the mode of propagation of the fungal prions.

## 1. Introduction

Prions are infectious proteins that can self-propagate and transmit their fibrillary amyloid conformation to normal indigenous prion proteins [1]. Prions can cause fatal neurodegenerative diseases that affect both humans and other animals. These diseases, called in general Transmissible Spongiform Encephalopathy (TSE), include Bovine Spongiform Encephalopathy (BSE or mad cow disease), Scrapie in sheep, Creutzfeldt–Jakob disease and Kuru in humans [1,2], etc. These diseases are caused by aggregation of the prion protein PrP in the amyloid form, which is a product of the *PRNP* gene in humans [3,4,5,6]. Besides mammalian prions, several prion-forming proteins were identified in fungi as well. While the biological significance of prion-forming proteins in fungi is somewhat unclear [7], most yeast prions are functional prions having a fibrillar structure similar to the mammalian prions [8], although, so far, there is no evidence for cross-nucleation of the mammalian prions with fungal prions. Thus, fungal prions have provided suitable and safe models for understanding the folding, aggregation and propagation mechanisms of disease-forming mammalian prions.

HET-s is a prion protein that corresponds to the [Het-s] prion system in the filamentous fungi *Podospora anserina* (*P. anserina*) [9]. It was initially identified as a part of a non-self-recognition process [10,11] that controls vegetative incompatibility in this fungus [12]. There are two antagonistic allelic variants of this protein: HET-s and HET-S, which are of the same length (289 amino acids) but differ in the sequence for 13 amino acids. The HET-s is a two-domain protein with a C-terminal domain (residues from 218 to 289) that is known as the prion-forming domain (PFD), and an α-helical globular N terminal domain (amino acid 1 to 227), also called the HeLo domain. The HeLo domain partially overlaps with the PFD [13,14,15]. The PFD of HET-s is composed of a two-fold repetition of 21 amino acids in an elementary triangular motif and adopts a β-solenoid structure with two layers of β-strands per monomer. Each of the repeated motifs composes one layer of the β-solenoid structure [16,17,18].

When a strain of the fungus carrying the HET-s prion encounters a strain that expresses a non-prion-forming form of the protein HET-S, the heterokaryon formed on cell fusion dies due to protein incompatibility [11]. Although the mechanism of cell death remains unclear, it is proposed that the incompatibility arises from the conversion of HET-S protein to β-solenoid rich prion conformation under influence of HET-s—the prion-forming form of the protein [14]. The structure of the PFD of HET-s has been solved by solid-state NMR [19]. Moreover, the correlation between structure and infectivity of HET-s has also been investigated by cryo-EM [20]. Thus, HET-s constitutes a good model system for studying prion aggregation and disaggregation.

It has been reported that ribosomes from bacterial, archaeal and eukaryotic sources can refold a large number of denatured proteins to their active state [21]. For all proteins tried so far, ribosome assisted folding showed a remarkably higher gain of activity compared to that achieved by spontaneous folding. The protein-folding activity of the ribosome (in short “PFAR”) [22,23,24,25,26] has been assigned to the large subunit of the ribosome, and more specifically, to the domain V of the main ribosomal RNA (rRNA; 23S in bacteria/25S in yeast/28S in higher eukaryotes) from the large subunit of the ribosome. Earlier reports have described PFAR as a specific target for two antiprion compounds 6-aminophenanthridine (6AP) and guanabenz acetate (GA) [27,28,29]. It has been shown that these compounds bind to the domain V of 23S/25S rRNA on specific nucleotides and competitively occludes interaction of the protein-folding substrates with domain V rRNA [28]. Since 6AP and GA reverse [PSI+] prion phenotype in vivo, it was proposed that PFAR is involved in prion processes [24,27,30]. Recently there are more reports for the involvement of PFAR in prion propagation in yeast [31,32].

The involvement of the nucleic acids in protein aggregation and disaggregation has been demonstrated with both DNA and RNA [33]. It has been shown using recombinant mammalian prion protein (rPrP) as a model system that nucleic acid interactions lead to different aggregated species of rPrP depending on the sequence and the size of the oligonucleotides [34]. Another earlier report claimed that prion formation in the budding yeast *Saccharomyces cerevisiae* (*S. cerevisiae*) was induced by the expression of rRNA from a plasmid-based rDNA construct [35]. Recent studies from our group showed that RNA can modulate aggregation of the core domain of the tumor suppressor protein p53 [36] and the rPrP [37]. Related results from other groups also showed that the *Escherichia coli* (*E. coli*) ribosome can prevent aggregation of partially folded protein intermediates [38]. However, so far, the role of rRNA in prion protein aggregation has remained elusive. In the present work, we have studied the role of various RNAs, including rRNA, mRNA and tRNA, in the aggregation of HET-s using Rayleigh light scattering and Thioflavin T (ThT) binding assays. We find that rRNA, especially the domain V of 23S/25S/28S rRNA, from various sources prevent the formation of large fibrillar aggregates of HET-s; this is also visualized with transmission electron microscopy (TEM). In contrast, “oligomeric seeds“ of HET-s are formed, which facilitate aggregation of fresh HET-s samples, as demonstrated with cross-seeding assays. Furthermore, using UV-crosslinking followed by primer extension assay we have identified HET-s interaction sites on the domain V of 23S rRNA and studied the effect of mutagenesis on the specific nucleotides. Our study demonstrates that rRNA-based PFAR can play an important role in modulating aggregation and propagation of the prions.

## 2. Results

### 2.1. rRNA Inhibits Aggregation of HET-s Prion as Monitored by Rayleigh Light Scattering

HET-s protein purified in urea denatured state was subjected to aggregation by incubation in 50 mM pH 7.5 Tris-HCl buffer overnight (8–12 h) at 37 °C and the extent of aggregation was followed by measuring Rayleigh light scattering at 402 nm. While freshly diluted HET-s showed only background level, a significant increase in light scattering was seen after overnight incubation of HET-s, indicating large aggregate formation in the HET-s sample. In order to study whether RNA and in particular rRNA influences HET-s aggregation, overnight incubation of HET-s was conducted in the presence of different RNA samples at a fixed concentration. These included bulk tRNAs isolated from *E. coli* MRE600, in vitro transcribed mRNAs from two unrelated proteins—(i) *E. coli* dihydrofolate reductase (DHFR) and (ii) human carbonic anhydrase I (HCA), and different domains of *E. coli* 23S rRNAs. Since full-length 23S rRNA produces high background scattering, individual domains of *E. coli* 23S rRNA were transcribed and subjected to this assay.

As shown in Figure 1A,B, addition of mRNAs or tRNAs did not show any significant change in light scattering suggesting that they do not influence HET-s aggregation. In contrast, a moderate to significant decrease in light scattering was seen with different domains of rRNAs. Domain V of 23S rRNA showed the highest reduction in light scattering, which suggests that this highly conserved rRNA domain strongly inhibits HET-s aggregation. Domains IV and II of 23S rRNA were also quite effective in reducing HET-s aggregation. It is worth mentioning that the domain V of 23S rRNA hosts peptidyl transferase center as well as the active sites for PFAR and the domains IV and II are closely associated with it.

Next, in vitro transcribed domain V rRNAs from large ribosomal subunit from various prokaryotic (*E. coli, Bacillus subtilis* (*B. subtilis*)) and eukaryotic (yeast *S. cerevisiae,* human, human mitochondria) sources were tested in HET-s aggregation assay. As shown in Figure 1C,D, all domain V rRNAs showed a comparable level of reduction in light scattering. This result suggests that in addition to peptidyl transfer and PFAR, prevention of protein aggregation is likely another conserved function of the domain V of 23S/25S/28S rRNA, irrespective of prokaryotic or eukaryotic origin. Further, upon titration of domain V rRNA (*E. coli*) gradual reduction of light scattering was observed in a concentration-dependent manner (Figure 1E,F). However, unlike PFAR, where a 1:1 molar ratio of the sample protein and domain V of rRNA is required for the highest extent of protein-folding, substoichiometric or relatively lesser concentration of domain V rRNA than the protein HET-s was sufficient for largest reduction of its aggregation. Our results indicate that rRNA especially the domain V of 23S rRNA can prevent the formation of large aggregates of the HET-s prion.

### 2.2. ThT Binding Demonstrates That rRNA Prevents Fibrillar Aggregation of HET-s

ThT binding assay is frequently used for the quantitative determination of amyloid fibril formation as its fluorescence increases specifically by binding to mature β-sheet enriched amyloid fibrils [39,40]. We have probed HET-s aggregation with ThT fluorescence under conditions identical to light scattering measurements. ThT was added just before fluorescence measurement, which excluded any effect of ThT on HET-s aggregation process.

As shown in Figure 2, ThT fluorescence increased significantly upon HET-s aggregation. This observation suggests that most likely HET-s aggregates to fibrillar amyloids, which is in good agreement with earlier reports [41]. Further, we analyzed the effect of the 23S rRNA domains (Domains II, IV, V of 23S rRNA from *E. coli*) in the ThT binding assay since these RNAs caused a significant reduction in light scattering by HET-s aggregates (Figure 2, inset). The bulk tRNAs isolated from *E. coli*, which did not have a pronounced effect in the light scattering assay was used as a control. Larger RNAs could not be used in this assay due to high background fluorescence of ThT with just RNAs. As expected, the bulk tRNAs did not reduce ThT fluorescence. However, all 23S rRNA domains lead to a decrease in ThT fluorescence (Figure 2, inset). Again, the highest reduction was seen with domain V of 23S rRNA. Moreover, the degree of reduction in ThT fluorescence with different 23S rRNA domains (Figure 2, inset) followed the same trend as seen in the light scattering assay (Figure 1E,F). This result confirms that rRNAs, especially the domain V of 23S rRNA, prevent amyloid aggregation of the HET-s prion.

### 2.3. HET-s Aggregates Show Different Morphology with or without RNAs as Seen by TEM

We have analyzed the aggregation morphology of HET-s proteins by TEM. As expected, after overnight aggregation reaction, the free HET-s protein forms typical amyloid fibrils (Figure 3A). Interestingly, when the aggregation reaction was done in the presence of domain V of 23S rRNA we have noticed, not only, a change in the fibril morphology, but also, a vast decrease in the amount of HET-s aggregation (Figure 3B). Our results showed that HET-s alone forms long amyloid fibrils as well as clustered aggregates from which the long fibrils emerge. In contrast, in the presence of domain V of 23S rRNA (*E. coli*), the fibrillar structures disappeared; instead, scattered smaller aggregates with spherical and branched structures were observed. Interestingly, RNAs could be seen as black dots among the HET-s aggregates as they bind better to uranyl ions supplied in the reaction as acetate salt. When other RNAs (e.g., DHFR mRNA, HCA mRNA) were tested with HET-s, various aggregate morphologies could be seen, which were different from the HET-s alone, but not free from fibrillar structures (Figure 3C,D). They caused accumulation of several intermediate aggregate structures, but the reduction of aggregation was not to the same extent as with domain V of 23S rRNA. This result visibly confirms that domain V rRNA effectively reduces and alters HET-s aggregation morphology.

### 2.4. Cross-Seeding Assay Demonstrates That Domain V of 23S rRNA Aids in Formation of The HET-s “Oligomeric Seeds“

To determine how domain V rRNA affects the HET-s fibril formation, aggregation kinetics of HET-s was followed by Rayleigh light scattering (Em. 402 nm, Ex. 400 nm) for HET-s alone or with domain V of 23S rRNA (*E. coli*; Figure 4). HET-s alone started aggregation immediately after dilution of the denaturant and light scattering increased with time suggesting gradual aggregation of the protein. When domain V rRNA was added with HET-s, both the rate and the amplitude of light scattering decreased dramatically (Figure 4), suggesting that significantly smaller amounts of large HET-s aggregates populated in the presence of the domain V rRNA. This result is in full agreement with end-point measurements presented in Figure 1A,B. However, to test whether the domain V rRNA aids in the formation of the oligomeric amyloid “seeds“, as also seen for murine rPrP [37], we designed a cross-seeding assay. For that, we first set HET-s aggregation reactions without and with domain V of 23S rRNA by incubating overnight at 37 °C. Then, the supernatant was collected after centrifuging down the large HET-s aggregates. A small volume of the cleared supernatant (10 μL) was added into a fresh HET-s aggregation reaction (500 μL) and then HET-s aggregation was followed with time using Rayleigh light scattering as described above. In both cases, we observed a pronounced and faster increase in light scattering as would be expected from seeded aggregation reactions (Figure 4). The addition of supernatant from the HET-s with domain V of 23S rRNA reaction showed the fastest and highest increase in light scattering suggesting that it contained “seeds” for HET-s aggregation. Thus, it could be concluded that domain V rRNA blocks the formation of large aggregates of HET-s, but facilitates the accumulation of small soluble aggregates. These can work as “seeds” to induce large aggregation in fresh HET-s solution.

### 2.5. The Interaction Map of HET-s on Domain V rRNA

Domain V of 23S rRNA was found to be an effective suppressor of fibrillary aggregation of HET-s. Here, we have mapped the interaction sites of HET-s with domain V rRNA using UV cross-linking followed by primer extension (by reverse transcription) assay. As shown in Figure 5, UV cross-linking of HET-s immediately after dilution of the denaturant produced reverse transcription “road-blocks” on almost the same nucleotides on domain V rRNA as other control protein substrates, namely HCA, DHFR and bovine carbonic anhydrase (BCA).. The main interaction sites were U2474–A2476, U2492–G2494, G2553–C2556, A2560–A2564 and U2585–G2588. Interestingly, the same sites were reported earlier for PFAR [28]. Thus, this observation suggests that PFAR might be involved in the prevention of HET-s aggregation. To test that we used a mutant variant of domain V rRNA and tested in HET-s aggregation assay.

### 2.6. Effect of Mutations in Domain V rRNA in HET-s Aggregation

HET-s protein produced a strong block in the residues UAG2586-88 (*E. coli* numbering) on domain V rRNA. These residues are highly conserved and mutation in these bases showed defects in PFAR. We tested the effect of UAG2586-88CCA mutant domain V rRNA on HET-s aggregation by light scattering and ThT binding assays. In both assays, UAG2586-88CCA domain V rRNA showed less efficiency in reducing HET-s aggregation compared to the wild-type domain V rRNA (Figure 6B,C), suggesting that interaction with these bases of domain V rRNA might have an impact on reduction of HET-s aggregation. However, it should be mentioned that PFAR was completely lost with this mutant domain V rRNA, while in case of HET-s aggregation it only caused partial reduction. Further, we tested this mutant domain V rRNA in UV cross-linking followed by primer extension assay. No binding was seen on the altered bases while other interaction sites remained unchanged (Figure 6A). This result suggests that HET-s interacts with domain V rRNA in a sequence-dependent manner. However, whether this interaction is directly causative for the reduction of HET-s aggregation or not remains to be answered.

## 3. Discussion

How newly synthesized polypeptide chains are folded in the living cells is one of the major questions in biological science. Several molecular chaperones were shown to be part of the process, which suggests that the cells have evolved multiple processes to ensure protein folding under various circumstances. Ribosomes from all three kingdoms of life were shown to have activity in refolding denatured proteins to their active state [21]. This PFAR commonly called PFAR was assigned to the large subunit of the ribosome, and more precisely to the domain V of the largest rRNA, which belongs to the large ribosomal subunit and holds the peptidyl transferase center [42]. A recent study demonstrated that ribosomes can also disaggregate various folding intermediates [38]. However, whether PFAR and protein disaggregation are related to two sides of the same coin remains elusive.

The interaction of the prion proteins with RNAs, resulting in modulation of their folding and aggregation pathways is an established fact [37,43,44]. As mentioned in the introduction, earlier results with the antiprion compounds 6AP and GA indicated a close involvement between PFAR and prion processes. These compounds were primarily identified by red/white screening in yeast [PSI+] system and further confirmed in the mammalian prion system [45,46]. The red/white colony screening method is based on the principle that in [PSI+] cells, most of the Sup35 protein, a subunit (also called eRF3) of the eukaryotic release factor, is sequestered into protein aggregates and thus unavailable to function in translation termination. As a result, [PSI+] causes an increased tendency to read through the stop codons. The *ade1-14* allele contains an opal stop codon in the open reading frame of ADE1. When Sup35p is in its aggregated, prion conformation ([PSI+] cells), ribosomes read through this opal codon, which allows cells to grow on adenine-deficient medium (SD-Ade) and produce regular white colonies. However, when Sup35p is in its normal soluble form ([PSI-] cells), translation of the *ade1-14* allele terminates at the opal codon preventing cells from growing on SD-Ade and leading to red colonies due to the formation of a metabolic byproduct. The treatment of [PSI+] cells with 6AP and GA results in the formation of red [PSI-] daughter colonies, suggesting that the prion phenotype was reversed. This would mean that either, in those 6AP/GA-treated cells, Sup35p could not aggregate to the inactive large amyloid form or alternatively, that prion propagation to daughter cells by means of small aggregates or “seeds“ formation was blocked. Since 6AP and GA bind specifically to rRNA and the binding is sensitive to mutations on domain V rRNA [28,46], PFAR was already implied in prion processes in vivo. However, the question remained whether PFAR is involved in “seeds” formation and thus in prion propagation, or alternatively—in large prion fibril formation.

In coherence with earlier reports [31,32], our results with HET-s prion protein as a model system, shed light on the involvement of the ribosome in the prion propagation processes. We find that rRNA, especially domain V of 23S/25S/28S rRNA, can prevent spontaneous aggregation of the HET-s prions into large, amyloid fibrils. Instead, it can facilitate the accumulation of the small oligomeric aggregates, which like prion “seeds”, can induce de novo fibrillar aggregation of HET-s. Our primer extension data presented in Figure 5 demonstrate that HET-s interacts with domain V of 23S rRNA using the nucleotides, which were identified for PFAR in relation to other nonprionogenic proteins [29,30,47]. Moreover, mutation of those nucleotides abolishes or diminishes the interaction (Figure 6A), also similar to what was seen earlier for other proteins [28]. This leads to the conclusion that HET-s interaction with rRNA is associated with PFAR. Thus, combining our observations together with earlier reports we propose that most likely, PFAR prevents misfolding of HET-s proteins and thereby blocks the formation of large, fibrillar and amorphous aggregates (Figure 2). However, PFAR does not inhibit HET-s aggregation completely leading to the formation of the oligomeric HET-s “seeds”. Our analyses are presented in a simple model in Figure 7. Our in vitro biochemical results can be extrapolated to explain the in vivo results of 6AP and GA action in [PSI+] yeast cells. In full agreement with the results and analyses presented by Voisset et al., we propose that PFAR is involved in the propagation of the [PSI+] prions by oligomeric “seeds” formation [32]. 6AP and GA primarily inhibit PFAR by binding to the domain V of 25S rRNA [28]. As a consequence, “seeds” formation diminishes and hence, prion propagation stops. Combining evidence from our current results and earlier works, we conclude that the rRNA-based PFAR governs yeast prion propagation by mediating a subtle balance between fibrillar (insoluble) and (soluble) oligomeric aggregates. The universality of this mechanism remains to be tested in other prion systems. However, given the highly conserved sequence, structure and functions of the domain V of the major rRNA of the large ribosomal subunit from all kingdoms of life, it will not be surprising if such a universal mechanism exists. This will, undoubtedly, be of fundamental scientific and therapeutic interest in the field of prion and neurodegenerative diseases.

## 4. Materials and Methods

### 4.1. Chemicals and Buffers for Experiments

The analytical grade chemicals were purchased from Sigma-Aldrich (Saint Louis, MO, USA) and Merck (Kenilworth, NJ, USA). Talon Resin (CLONTECH) was purchased from TaKaRa Bio Europe AB (Göteborg, Sweden). The reagents for in vitro transcription, primer extension assay and extraction of RNAs were purchased from Macherey-Nagel (Dueren, Germany) and ThermoFisher Scientific (Uppsala, Sweden).

### 4.2. HET-s Protein Expression and Purification

The pET21 clone of full-length HET-s with C-terminal histidine-tag was kindly provided by Sven J. Saupe (University of Bordeaux, Bordeaux, Aquitaine, France). The plasmid was transformed into *E. coli* BL21(DE3) pLysS cells. Bacteria were grown to 0.5 OD in 2× YT medium and then induced by the addition of 1 mM isopropyl-β-d-thiogalactoside. Four hours after induction, the cells were harvested by centrifugation and either stored at −80 °C or proceeded with purification. Cells were lysed in lysis buffer (100 mM potassium phosphate buffer, pH 8.0). The lysate was centrifuged for 20 min at 20,000× *g*. The pellet was washed in lysis buffer and resuspended in denaturing buffer (8 M guanidinium-HCl (Gdn-HCl) in lysis buffer). The lysate was incubated with Talon Resin (CLONTECH) for 1 h at 20 °C, and the resin was washed with washing buffer (8 M urea in lysis buffer). The HET-s protein was eluted from the resin in the denatured state with elution buffer (200 mM imidazole in washing buffer) and stored at 4 °C.

### 4.3. In Vitro Transcription and Extraction of Various RNAs

Plasmids containing DNA sequences for domain V of large rRNA from different species (Human, Human mitochondria, bacteria *B. subtilis*, yeast *S. cerevisiae*) and PCR products containing sequences of 23S rDNA from *E. coli*, mRNAs of HCA (783 nucleotides (nt)), DHFR (498 nt) and were used as DNA templates for transcription. The in vitro transcriptions were done using T7 RNA polymerase according to [30] and the RNAs were purified from free nucleotides by using RNA purification kit (Macherey-Nagel). Bulk tRNAs were isolated from *E. coli* MRE600 by phenol-chloroform treatment [48]. The quality of the RNAs was checked by running into denaturing urea polyacrylamide gel. The length of the 23S rRNA domains were as domain V (595 nt), domain IV (360 nt) and domain II (725 nt).

### 4.4. Light Scattering Assay

Rayleigh light scattering is often used to monitor protein aggregation since the intensity of the scattered light increases with the increase in the size and density of the particles. For studying HET-s aggregation, 8 M urea denatured HET-s was diluted 50 times in 50 mM Tris-HCl buffer (pH 7.5) to a final concentration of 5 µM and incubated without or with different RNA samples overnight at 37 °C. Then, Rayleigh light scattering from the samples was measured at 402 nm (excitation 400 nm, excitation and emission slit 2.5 nm) with a HITACHI F-7000 steady-state fluorescence spectrophotometer (Tokyo, Japan) at 25 °C. For kinetics of HET-s aggregation light scattering at 402 nm was followed with time with the same setup as described above, alone or with various RNAs/“seeds“ from previous HET-s aggregation reactions. All measurements were performed at least in triplicates and the data represent the average of three to five independent experiments.

### 4.5. ThT Binding Assay

The HET-s samples with/without RNAs were treated in the same way as in the light scattering assay. ThT (obtained from Sigma) solutions were prepared in double-distilled water and filtered through a 0.22 μm syringe filter. To the overnight incubated HET-s samples ThT was added in the ThT:HET-s ratio 20:1 incubated 3 min at 25 °C, and then ThT fluorescence (between 465 and 565 nm, excitation and emission slit width 5 nm) was recorded using a fluorescence spectrophotometer (HITACHI, F-7000) with excitation at 450 nm. All measurements were done in triplicates and the data represent the average of all three experiments after background subtraction.

### 4.6. Primer Extension Assay for Detecting HET-s Binding Sites on Domain V rRNA

30 μM HET-s protein stored in urea was diluted 100 times in refolding buffer containing domain V variants of 23S rRNA from *E. coli* (300 nM), and UV cross-linking was performed immediately in a Bio-Rad GS Gene Linker TM instrument (Hercules, CA, USA), with 254 nm UV irradiation (600 mJ) [49]. For comparison, three unrelated proteins—BCA, HCA and DHFR, were denatured with 6 M Gdn-HCl and subjected to UV cross-linking immediately after dilution of the denaturant. The samples were kept on ice during irradiation to prevent heat damage to the RNA. The irradiated samples were precipitated by salt/ethanol and washed with 70% ethanol for primer extension. Primer 5′-ACCCCGGATCCGCGCCCACGGCAGATAGG-3′ was labeled with [γ-32P] dATP at 37 °C using T4 polynucleotide kinase for 1 h by the 5′-end-labeling method [50]. The primer extension was done using the same procedure as described in [28].

### 4.7. TEM

HET-s protein stored in urea was diluted to 5 μM and incubated in 50 mM Tris-HCl pH 7.5 at 37 °C for a day without or with 1 μM in vitro transcribed different RNAs. For morphological analysis of aggregates formed in vitro, samples were diluted 1:4 in 50 mM pH 7.5 Tris-HCl. A solution of each sample (10 μL) was applied to a carbon-coated copper grid and negatively contrasted with 2.5% uranyl acetate in 50% ethanol. Samples were studied at 75 kV in a Hitachi H-7100 transmission electron microscope (Tokyo, Japan), and images were obtained with Gatan 832 Orius SC1000 (Gatan Inc., Pleasanton, CA, USA).

### 4.8. Cross-Seeding Assay

First, 8 M urea denatured HET-s was diluted 50 times in 50 mM Tris-HCl buffer (pH 7.5) to a final concentration of 5 µM and incubated without or with domain V of 23S rRNA (1 μM) overnight at 37 °C to induce aggregation. The overnight samples were centrifuged at 14,000 rpm for 30 min at room temperature and the supernatant was separated from the aggregated pellet. 10 μL of the cleared supernatant from each reaction was added as “seeds” to the fresh dilutions of 8 M urea denatured HET-s (500 μL). The aggregation kinetics was followed by monitoring Rayleigh light scattering at 402 nm (excitation 400 nm, excitation and emission slit 2.5 nm) with a HITACHI F-7000 steady-state fluorescence spectrophotometer as described under “Light Scattering Assay”, we added as controls, we also followed the kinetics of HET-s aggregation with and without domain V of 23S rRNA (1 μM). The fluorescence data are plotted against time to follow the time course of HET-s aggregation with/without “seeds“. All experiments were done in triplicates.

## Figures and Tables

**Figure 1 ijms-21-06340-f001:**
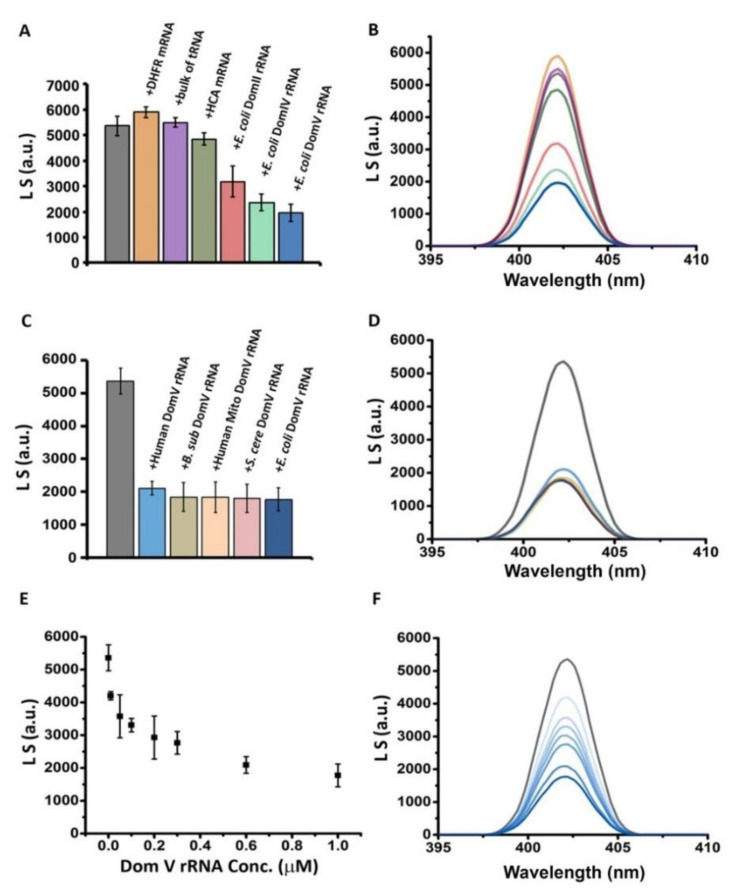
The effect of RNAs in HET-s aggregation as followed by Rayleigh light scattering at 402 nm. HET-s protein stored in a denatured condition in 8 M urea was diluted in 50 mM Tris-HCl pH 7.5 to a final concentration of 5 µM and incubated without or with different RNA samples (~1 µM) overnight at 37 °C for aggregation. Then, light scattering was measured with a HITACHI F-7000 fluorescence spectrophotometer. While the left panels show emission peak intensities at 402 nm (excitation 400 nm; average of three to five independent experiments, error bars represent standard deviation), the right panels show representative spectra using the same color codes as in the corresponding left panels. Light scattering from HET-s (**A**,**B**) without or with different RNA samples, (**C**,**D**) with domain V of rRNA from various sources and (**E**,**F**) with different concentrations of domain V rRNA from *E. coli*.

**Figure 2 ijms-21-06340-f002:**
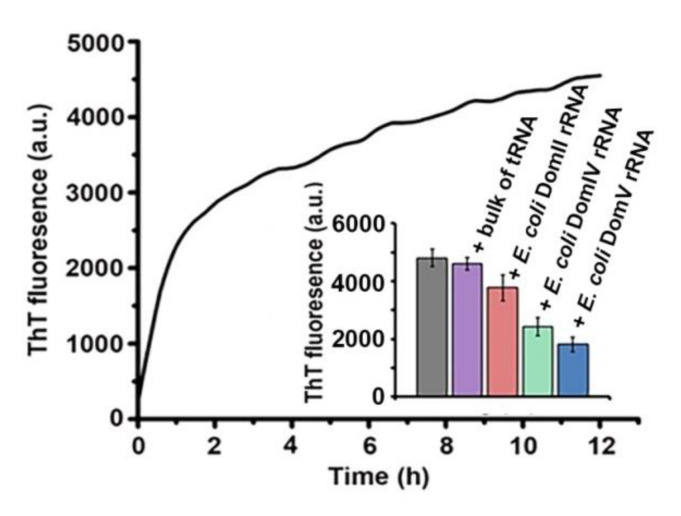
The effect of different RNAs on HET-s aggregation monitored by ThT binding. HET-s aggregation was induced as mentioned in the Materials and Methods section and tested for ThT binding. The curve shows resulting ThT fluorescence (Emission 483 nm, excitation at 450 nm) with time. The inset represents the saturating values of ThT fluorescence (at 483 nM) without (gray bar) or with various RNAs as indicated. The data presented here are average of three independent measurements and the error bars represent standard deviation.

**Figure 3 ijms-21-06340-f003:**
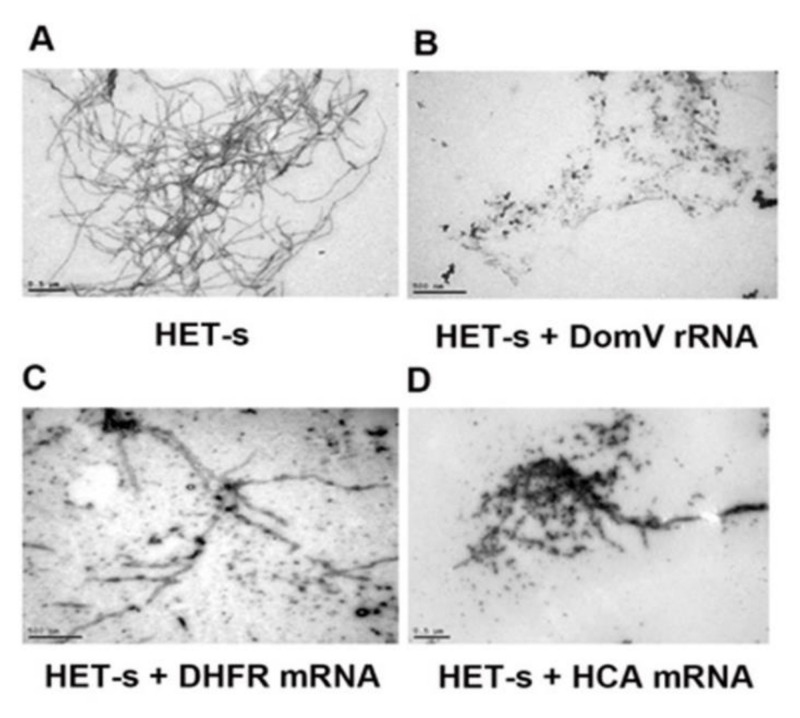
TEM images of HET-s without (**A**) or with (**B**–**D**) different RNAs. HET-s protein stored in urea was diluted to 5 μM and incubated in 50 mM Tris-HCl pH 7.5 at 37 °C for a day without or with 1 μM in vitro transcribed domain V of *E. coli* 23S rRNA (**B**), DHFR mRNA (**C**) and HCA mRNA (**D**). Then, TEM images were produced using a Hitachi H-7100 transmission electron microscope (Hitachi, Tokyo, Japan).

**Figure 4 ijms-21-06340-f004:**
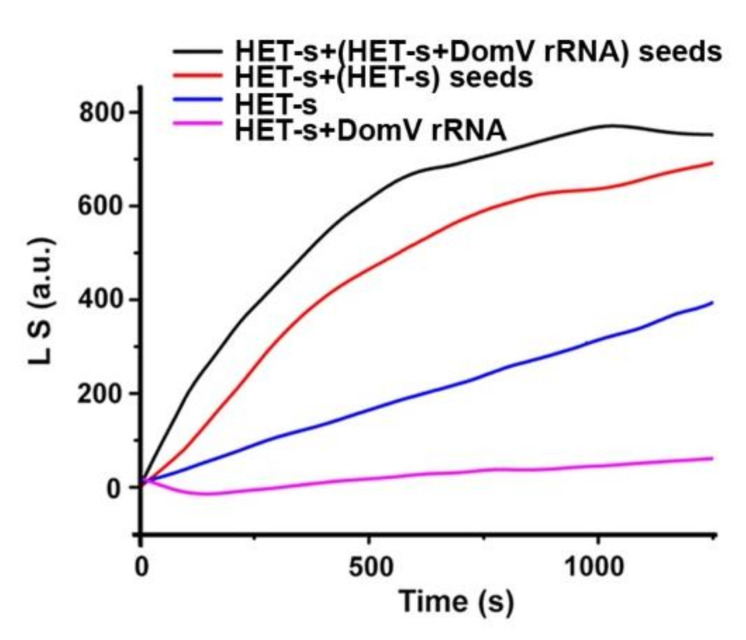
Aggregation kinetics of HET-s with “seeds” from matured HET-s aggregation reactions, conducted with or without domain V of 23S rRNA The aggregation kinetics is followed by Rayleigh light scattering at 402 nm (excitation 400 nm), without (blue trace) or with domain V of 23S rRNA (1 μM; pink trace) and with “seeds” from previous HET-s aggregation reactions, conducted without or with domain V of 23S rRNA. See Materials and methods for details. The traces in the figure are representative out of three independent replicates.

**Figure 5 ijms-21-06340-f005:**
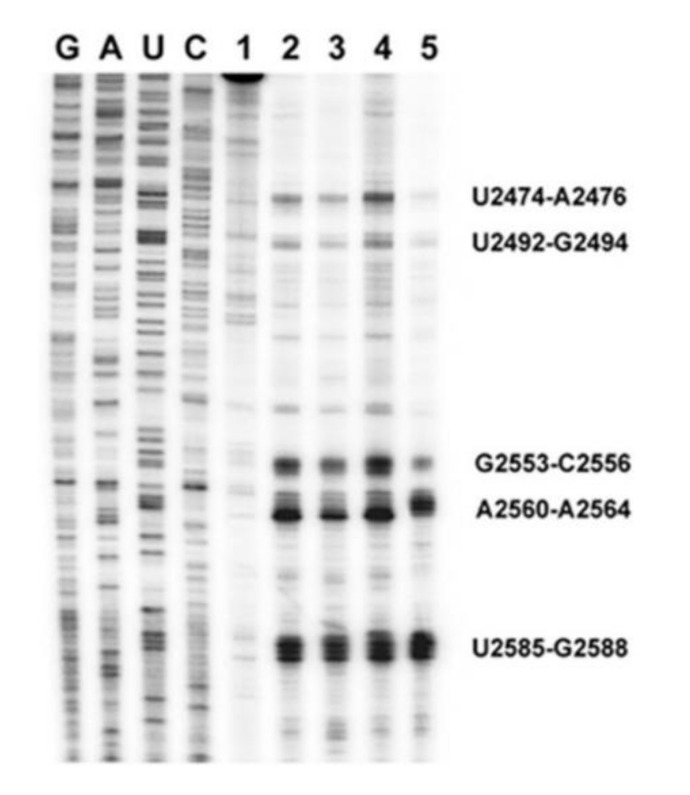
Primer extension or “road-block” analysis of domain V of 23S rRNA after UV cross-linking of HET-s and other protein while refolding. HET-s and three other protein controls were subjected to UV cross-linking with domain V of 23S rRNA *E. coli* immediately after dilution from the denaturant as mentioned in the Materials and Methods. Then, primer extension was performed using reverse transcriptase and the products were run on a sequencing gel—without any protein (**Lane 1**), with HCA (**Lane 2**), DHFR (**Lane 3**), BCA (**Lane 4**) and HET-s (**Lane 5**). The first four lanes in the left show sequencing ladders as indicated on top. The road-block sites are labeled with nucleotide numbers corresponding to *E. coli* 23S rRNA. Part of this gel was originally published in the Journal of Biological Chemistry [28]. © the American Society for Biochemistry and Molecular Biology.

**Figure 6 ijms-21-06340-f006:**
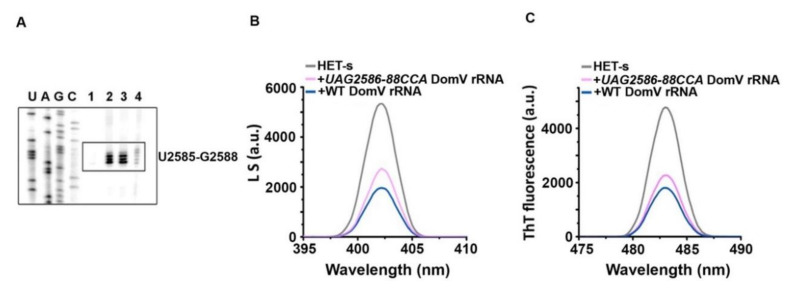
The effect of UAG2586–88CCA mutation in domain V rRNA on HET-s aggregation and interaction map**.** (**A**) The primer extension map of sections of domain V of 23S rRNA (*E. coli)* without any protein (**Lane 1**) or with HET-s UV cross-linked to wild-type (**Lane 2**), or mutant domain V rRNA with mutations UU2561–62AA (**Lane 3**) and UAG2586-88CCA (**Lane 4**). Only the road-block site U2585-G2588 is shown in the box; (**B**,**C**) HET-s aggregation with UAG2586-88CCA mutant variant of domain V rRNA checked by light scattering (**B**) and ThT binding (**C**) using the setting described in materials and methods. Figures are representative of three independent triplicates.

**Figure 7 ijms-21-06340-f007:**
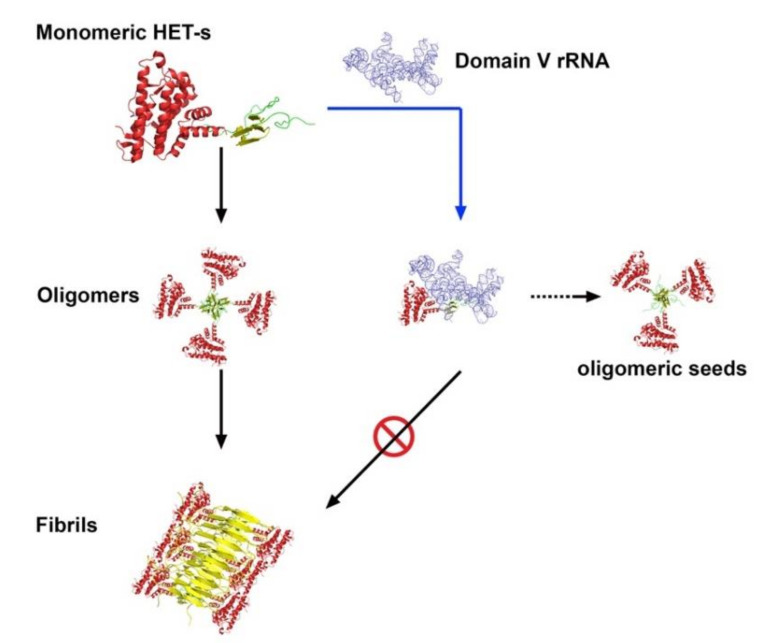
Schematic illustration of the HET-s aggregation pathway and the role of domain V of 23S rRNA. Monomeric HET-s spontaneously undergoes a conformational change to form prefibrillar oligomers, which eventually aggregate to large insoluble amyloid fibrils. However, when it interacts with domain V of 23S rRNA (or other active rRNA components) PFAR prevents misfolding of HET-s and thereby blocks the formation of large, fibrillar (and amorphous) aggregates. Instead, an alternative pathway comes into action and HET-s folds to form soluble “oligomeric seeds” capable of promoting de novo prion propagation.

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
