# Peer review of "Ribosomal RNA Modulates Aggregation of the *Podospora* Prion Protein HET-s"

_ijms, 2020, doi:10.3390/ijms21176340_

Round 1

Reviewer 1 Report

I am satisfied with the changes made by the authors.

Reviewer 2 Report

The authors implemented the manuscript following the indications given. The revised manuscript is acceptable for publication at this stage.

This manuscript is a resubmission of an earlier submission. The following is a list of the peer review reports and author responses from that submission.

Round 1

Reviewer 1 Report

Sanyal and coworkers examine the effect of various RNAs on in vivo aggregation of a protein HET-s from Podospora anserina. Several rRNA fragments have a stronger effect on aggregation that other teste RNAs. The authors conclude that rRNA affect propagation of prions.

Critique:

This study makes dramatic claims that are made on the bases of poorly controlled experiments. Therefore, in my view this work is unacceptable for publication.

Authors answer:

We accept the critical comments from the reviewer. Since we and other groups have demonstrated the role of rRNA in protein folding and aggregation in many papers published earlier (see references 1 – 6), we have not demonstrated all controls in this particular work. At the same time, controls presented in some crucial experiments (e.g. Figure 4), have escaped the attention of the reviewer. We have supplied the control data for the reviewer wherever we feel appropriate.

Main points:

2.1. I have serious doubts that fairly long RNA would ‘survive’ an overnight incubation at 37°C. Furthermore, different RNA would likely have different stability. Therefore, the observed effects could be an indirect consequence of differential stability of RNA under the conditions of in vitro experiments, rather than reflect any functionally-meaningful effects pertaining to what could be happening in the cell. Without demonstrating that RNA remains intact (or that at least the same amount of different RNAs remain intact) at the end of the overnight incubation, the experiments is impossible to interpret in any unambiguous form.

Authors answer:

We agree with the reviewer that different RNAs may have different stability. However, under our minimal experimental setup, where experiments are conducted very carefully with the addition of the RNAse inhibitors, the RNAs are not degraded during the experimental time. We add a gel image (Figure 1) here for the two RNAs used in this work after incubation with HET-s for eight hours at room temperature. As a control, we add another mRNA as well. The smaller faint band in the domain V rRNA is 5S rRNA, which has not been fully separated from the 23S rRNA. The image clearly shows that the RNAs are not degraded during the experimental incubation.

image_2020_09_02T08_02_06_384Z.png

Being said that, we will also like to emphasize that the intact RNA is not needed for the entire time (overnight 8 – 12 hours) of incubation for the steady state HET-s aggregation. This is evident from the (new) Figure 4 of the manuscript, which shows the aggregation kinetics of HET-s alone and also with domain V of 23S rRNA. The kinetics shows that the HET-s aggregation saturates in about 1000 sec. Thus, whether or not the RNAs remain intact for the entire incubation period for the steady-state measurements is not relevant.

2.4. and Figure 4. The crosslinking experiment lacks the critical control: primer extension on rRNA that was incubated with the proteins without crosslinking. There is a good chance that the observed ‘roadblock’ bands result from nuclease cleavage of RNA rather than from UV crosslinking, moreover that the same set of bands is seen with very diverse proteins.

Authors answer:

The lane 1 in the gel presented at Figure 4 is the control – domain V of 23S rRNA without protein. This RNA has been treated exactly in the same way as the ones with proteins. The lack of the road-blocks in this control lane and the similar road-block positions with various proteins clearly demonstrate that the bands in the lanes 2 – 5 do not result from nuclease cleaving. The primer extension experiments are repeated many times with highly reproducible results. We
can provide multiple gel images if needed.

For information, the same road-block experiment has been done many times earlier to demonstrate the exact interaction sites of the proteins on RNA (see references 10. 11, 17, 18). In fact, the road-block technique was developed in Harry Noller’s lab (Ref 18), where too rRNAs were employed. We request the reviewer to go thorugh those papers. The earlier demonstrations together with the control in lane 1, we demonstrate that the complaint from the reviewer regarding appearance of the bands in lanes 2 – 5 due to nuclease cleavage is baseless.

Shockingly, The gel shown in Fig. 4 is EXACTLY THE SAME GEL (!!!) that was published in the 2013 JBC paper by Sanyal group with addition of one extra lane (compare with Fig. 2A in the 2013 paper). This is highly irregular, to say the least!

Authors answer:

We agree completely that part of this gel (till lane 4) has been published in our previous JBC paper, 2013. We did ask permission from JBC for reuse of the image in this publication. In response to that the JBC editorial assistant pointed to us the JBC policy that the authors have full right to use the images published in JBC for their future publications. We provided web link below. We have done one careless mistake though. We have not mentioned it clearly in the Figure Legend. The mistake has been corrected now and we have referred to the JBC paper following the statements from the ASBMB site https://www.asbmb.org/journals-news/editorial-policies#copyright.

2.4. Aren't the authors surprised that changing only two nucleotides in domain V of the 23S rRNA has such a dramatic effect on HET-s aggregation whereas replacing the entire domain V with a completely different RNA sequence (e.g. domain VI of 23S rRNA) has only a minor effect (Figs. 1A and 3B)?

Authors answer:

We will like to guide the reviewer to the following papers where mutation in single nucleotide has shown change in the interaction pattern of the RNA with different sample proteins. (references 10, 12). Thus, this result is not surprising to us or to any experienced RNA biologist. Instead, we strongly claim that the loss of road-blocks by point mutation in the specific residues of 23S rRNA indicates that those sites are in fact the sites for interaction of the protein. Similar experiment and results have been published for other proteins such as pig muscle lactate dehydrogenase and pig heart cytoplasmic malate dehydrogenase (reference 14), beta-galactosidase (reference 15), Cytoplasmic Malate Dehydrogenase (reference 16) etc..
Mutational scanning has been instrumental in Molecular Biology research to demonstrate the role of specific nucleotides in innumerous studies. We are in fact surprised by this comment from the knowledgeable referee.
When it comes to domain IV, there must be some key nucleotides as well for its role in the process, which we have not investigated in the current study.

Other points:

2.1. What does ‘overnight incubation” mean? 8 h? 12h? 16h?

Authors answer:

8 – 12 hour. This has now been added in the Materials and Methods.

2.2. Even if domain V of the 232S rRNA reduces aggregation, it does not mean that it is “A conserved function of domain V rRNA”. High concentration of NaCl will also prevent aggregation, but it does not mean that this is ‘function of NaCl’!

Authors answer:

We understand the comment and accept the criticism. We have rephrased the sentence as the following:
prevention of protein aggregation is another function of the domain V of 23S / 25S / 28S rRNA’.

2.2. (“These results suggest that inhibition of HET-s aggregation can be attributed specifically to ribosomal RNA”). The correct statement would be that ‘in our experiments, it was rRNA that inhibited HET-s aggregation’. The fact that three other tested RNAs had lesser effect does not mean that it is a SPECIFIC effect of rRNA.

Authors answer:

We understand the point raised about ‘specificity’ and rephrased the text as suggested.
‘These results suggest that rRNA promotes inhibition of HET-s aggregation under our experimental
condition.’

2.3.1 and Fig. 3a. The ‘other RNA’ control is missing. The size of the dot may be affected by the absorption properties of protein on the filter in the presence or absence of added RNA rather than reflect the amounts of amyloid oligomers.

Authors answer:

We understand the comment from the reviewer. We have published dot-blot results with the same antibody and the same assay in our earlier publication Kovachev et al., J. Biol. Chem. (2017) 292(22) 9345–9357, where other RNAs are used too. But, we did not have dot-blot data with HET-s and other RNAs as this experiment was conducted with the focus on the domain V of 23S rRNA alone. We aimed to repeat the experiment with other RNAs. But, due to long time gap and situations due to Covid-19, the antibody could not be purchased and the experiment could not be repeated. Thus, as a benefit of doubt, we exclude the dot-blot experiment from the revised manuscript.

3. The first two paragraph of the discussion belong to Introduction. They discuss only the previous experiments not the ones obtained in this study.

Authors answer:

We feel that it is appropriate to introduce the topic briefly in the beginning of the ‘discussion’ and this is helpful for the new and the neutral readers. From my 20 years of academic career as an independent researcher, PhD and postdoc supervisor, reviewer and editor of high impact journals, I can assure that this is not uncommon! At least, this cannot be the basis of rejection of this manuscript.

4.2. The reference to the protocol used for T7 transcription is apparently missing.

Authors answer:

Thanks to the reviewer for pointing it out. It has been added to the manuscript.

Reviewer 2 Report

In this manuscript, the authors present compelling evidence that specific regions of the rRNA stimulate formation of oligomeric seeds of the prion protein HET-s while also inhibiting the formation of amyloid aggregates. They demonstrate their hypothesis using a range of techniques to support previous obversations and expand the understanding of prion aggregation. This information is well presented and should further the aims of treating prion diseases. I can recommend this paper for publication, but would suggest that the authors consider my comments below to improve the clarity of the manuscript.

Here are some minor comments about the paper:

1. In the first paragraph of section 2.2, the authors refer to stoichimetric and sub-stoichimetric amounts of rRNA. Could the authors state what range of concentrations they mean in each instance?

Authors answer:

The stoichiometric amount means 1: 1 molar ratio (protein : RNA) with concentration in the sub-micromolar (300 nM) to low micromolar range (5 μM). The sub-stoichiometric concentration means that the RNA concentration is less than protein concentration. However, the actual concentration of the sample protein and the rRNA is not important in the context of the sentence. Rather their relative ratio is important. For clarity, we have modified the sentence as:
‘However, unlike PFAR, where 1: 1 molar ratio of the sample protein and the domain V of rRNA is required for obtaining the highest extent of protein folding, sub-stoichiometric or relatively lesser concentration of domain V rRNA is enough for largest reduction of HET-s aggregation’.

2. I’m not sure why section 2.3 is presented as sub-headings. It would be just as clear if made into normal paragraphs with an introductory sentence for each, and the article would flow better.

Authors answer:

We agree with the reviewer’s suggestion. In fact, after careful revision we decide to move the ‘Thioflavin T binding’ results immediately after ‘Rayleigh Light Scattering’ results in 2.2. We have removed the dot-blot results (see full reason below). Thus, we added the cross-seeding assay results in a separate heading 2.4.

3. Figure 3C is a little confusing since it isn't immediately clear that the black line refers to HET-s protein that has been seeded with the supernatant incubated with rRNA. As it is written, it appears that rRNA is present in the second reaction as well, which would be predicted to give results similar to the pink line. Perhaps the labelling could be something like shown below. If the authors don't think that is any clearer (I'm not sure myself), perhaps a flow diagram could be inset into the figure.

    • HET-s + (HET-s + Dom V rRNA) seeds
    • HET-s + (HET-s) seeds
    • HET-s
    • HET-s + Dom V rRNA

Authors answer:

We thank the reviewer for pointing it out. We have updated the figure labelling following the suggestion, which has certainly improved it.

4. For Figure 3A, how many times was this experiment performed and could the experiment be quantitated and presented as a bar graph? Indeed, there are a few instances (3A, 3C, 5B-C) where it would be comforting to mention the n numbers in the figure legends to show that the experiments are reproducible, even if specific statistical tests are not presented.

Authors answer:

We have already mentioned in all figure legends the number of replicates (typically three). The number of replicates is also added in the Materials and Methods for all experiments.

5. The graphs are often quite low resolution with very small text.

Authors answer:

We have separate high resolution images. However, given the format of the journal, we have improved all
Figures and placed in the text accordingly.

Reviewer 3 Report

The manuscript entitled “Ribosomal RNA Modulates Aggregation of the Podospora Prion Protein HET-s” by Pang et al., reports the studies in vitro of the possible role as inhibitor of aggregation processes by domain V rRNA. In fact, the authors after studies on various RNAs (mRNA, tRNA and rRNA) have focused on domain V rRNA. Many techniques have been used to follow the process and to give a possible model capable of explaining the inhibiting properties of domain V rRNA.

In this framework, these studies provide novel information regarding role of the nucleic acids in prion aggregation/disaggregation.

Authors answer:

We are thankful to the reviewer for proper summary of our paper and also for understanding that we are
reporting a novel role of RNA in relation to prion aggregation / disaggregation.

However, several points must be revised to improve the quality of the manuscript.

Major revisions:

1. The homogeneity of HET-s protein is very important in these experiments. The Authors must provide analytical evidence of purity (e.g. SDS-PAGE).

Authors answer:

We attach below a gel for HET-s in different dilutions to demonstrate the homogeneity of HET-s protein purification. We however feel that it is not necessary to add this gel in the manuscript. We have written that the protein was purified with >99 % purity.

2.PNG

2. The Authors claim to have used tRNA but give no information on this RNA. This information should be added to the methods.

Authors answer:

Please see the RNA gels added. Since we have published many works with these RNA domains earlier, we do not think that it is necessary to increase the length of the paper by adding the RNA gels.

3. The Authors used domain V rRNA of various species. In the results or as supplemental materials, a gel of these rRNAs should be placed to show the integrity of the domains used.

Authors answer:

We thank the reviewer for pointing it out. The earlier Figure 6 is now Figure 7. We have removed a and b and rephrased the legend.

4. In the legend of figure 6 what do "a" and "b" indicate? there are no in this figure.

Authors answer:

We thank the reviewer for pointing it out. The earlier Figure 6 is now Figure 7. We have removed a and b and rephrased the legend.

Finally, “in vitro” it must be reported in italics.

Authors answer:

Changed as suggested.

References used in this response letter

1. Kudlicki, W.; Coffman, A.; Kramer, G.; Hardesty, B., Ribosomes and ribosomal RNA as chaperones for folding of proteins. Fold Des 1997, 2, (2), 101-8.
2. Tribouillard-Tanvier, D.; Dos Reis, S.; Gug, F.; Voisset, C.; Beringue, V.; Sabate, R.; Kikovska, E.; Talarek, N.; Bach, S.; Huang, C.; Desban, N.; Saupe, S. J.; Supattapone, S.; Thuret, J. Y.; Chedin, S.; Vilette, D.; Galons, H.; Sanyal, S.; Blondel, M., Protein folding activity of ribosomal RNA is a selective target of two unrelated antiprion drugs. PLoS One 2008, 3, (5), e2174.
3. Blondel, M.; Soubigou, F.; Evrard, J.; Nguyen, P. H.; Hasin, N.; Chedin, S.; Gillet, R.; Contesse, M. A.; Friocourt, G.; Stahl, G.; Jones, G. W.; Voisset, C., Protein Folding Activity of the Ribosome is involved in Yeast Prion Propagation. Sci Rep 2016, 6, 32117.
4. Voisset, C.; Blondel, M.; Jones, G. W.; Friocourt, G.; Stahl, G.; Chedin, S.; Beringue, V.; Gillet, R., The double life of the ribosome: When its protein folding activity supports prion propagation. Prion 2017, 11, (2), 89-97.
5. Cordeiro, Y.; Vieira, T.; Kovachev, P. S.; Sanyal, S.; Silva, J. L., Modulation of p53 and prion protein aggregation by RNA. Biochim Biophys Acta Proteins Proteom 2019, 1867, (10), 933-940. 6. Kovachev, P. S.; Gomes, M. P. B.; Cordeiro, Y.; Ferreira, N. C.; Valadao, L. P. F.; Ascari, L. M.; Rangel, L. P.; Silva, J. L.; Sanyal, S., RNA modulates aggregation of the recombinant mammalian prion protein by direct interaction. Sci Rep 2019, 9, (1), 12406.
7. Garcia-Jove Navarro, M., Kashida, S., Chouaib, R. et al. RNA is a critical element for the sizing and the composition of phase-separated RNA–protein condensates. Nat Commun 10, 3230 (2019).
8. Horowitz S, Bardwell JC. RNAs as chaperones. RNA Biol. 2016;13(12):1228-1231.
9. Saha S, Hyman AA. RNA gets in phase. J Cell Biol. 2017;216(8):2235-2237.
10. Pang, Y.; Kurella, S.; Voisset, C.; Samanta, D.; Banerjee, D.; Schabe, A.; Das Gupta, C.; Galons, H.; Blondel, M.; Sanyal, S., The antiprion compound 6-aminophenanthridine inhibits the protein folding activity of the ribosome by direct competition. J Biol Chem 2013, 288, (26), 19081-9.
11. Das D, Samanta D, Hasan S, et al. Identical RNA-protein interactions in vivo and in vitro and a scheme of folding the newly synthesized proteins by ribosomes. J Biol Chem. 2012;287(44):37508-37521.
12. Chowdhury S, Pal S, Ghosh J, DasGupta C. Mutations in domain V of the 23S ribosomal RNA of Bacillus subtilis that inactivate its protein folding property in vitro. Nucleic Acids Research. 2002 Mar;30(5):1278-1285
13. Das B, Chattopadhyay S, Das Gupta C. Reactivation of denatured fungal glucose 6-phosphate dehydrogenase and E. coli alkaline phosphatase with E. coli ribosome. Biochem Biophys Res Commun. 1992;183(2):774-780.
14. Pal D, Chattopadhyay S, Chandra S, Sarkar D, Chakraborty A, Das Gupta C. Reactivation of denatured proteins by domain V of bacterial 23S rRNA. Nucleic Acids Res. 1997;25(24):5047-5051.
15. Chattopadhyay S, Pal S, Pal D, Sarkar D, Chandra S, Das Gupta C. Protein folding in Escherichia coli: role of 23S ribosomal RNA.
16. Sanyal SC, Pal S, Chowdhury S, DasGupta C. 23S rRNA assisted folding of cytoplasmic malate dehydrogenase is distinctly different from its self-folding [published correction appears in Nucleic Acids Res. 2002 Dec 15;30(24):5593.. Chaudhuri Saheli [corrected to Chowdhury Saheli]]. Nucleic Acids Res. 2002;30(11):2390-2397.
17. Samaha RR, Joseph S, O'Brien B, O'Brien TW, Noller HF. Site-directed hydroxyl radical probing of 30S ribosomal subunits by using Fe(II) tethered to an interruption in the 16S rRNA chain. Proc Natl Acad Sci U S A. 1999;96(2):366-370.
18. Holmberg L, Noller HF. Mapping the ribosomal RNA neighborhood of protein L11 by directed hydroxyl radical probing. J Mol Biol. 1999;289(2):223-233.